# Role of Interferons in *Mycobacterium tuberculosis* Infection

**Gaithrri Shanmuganathan** [1], **Davit Orujyan** [1], **William Narinyan** [1], **Nicole Poladian** [1], **Sanya Dhama** [2], **Arpitha Parthasarathy** [2], **Alexandra Ha** [1], **Daniel Tran** [1], **Prathosh Velpuri** [1], **Kevin H. Nguyen** [1] **and Vishwanath Venketaraman** [1,*]

[1] College of Osteopathic Medicine of the Pacific, Western University of Health Sciences, Pomona, CA 91766, USA

[2] Keck Science Department, Pitzer College, Claremont, CA 91711, USA

* Correspondence: vvenketaraman@westernu.edu; Tel.: +1-909-706-3736

**Abstract:** Considerable measures have been implemented in healthcare institutions to screen for and treat tuberculosis (TB) in developed countries; however, in low- and middle-income countries, many individuals still suffer from TB's deleterious effects. TB is caused by an infection from the *Mycobacterium tuberculosis* (*M. tb*) bacteria. Symptoms of TB may range from an asymptomatic latent-phase affecting the pulmonary tract to a devastating active and disseminated stage that can cause central nervous system demise, musculoskeletal impairments, and genitourinary compromise. Following *M. tb* infection, cytokines such as interferons (IFNs) are released as part of the host immune response. Three main classes of IFNs prevalent during the immune defense include: type I IFN (α and β), type II IFN (IFN-γ), and type III IFN (IFN-λ). The current literature reports that type I IFN plays a role in diminishing the host defense against *M. tb* by attenuating T-cell activation. In opposition, T-cell activation drives type II IFN release, which is the primary cytokine mediating protection from *M. tb* by stimulating macrophages and their oxidative defense mechanisms. Type III IFN has a subsidiary part in improving the Th1 response for host cell protection against *M. tb*. Based on the current evidence available, our group aims to summarize the role that each IFN serves in TB within this literature review.

**Keywords:** type I IFN; type II IFN; type III IFN; M. tb; IFN-α; IFN-γ; IFN-λ; IFN β

## 1. Introduction

Tuberculosis (TB) is caused by the *Mycobacterium tuberculosis* (*M. tb*) bacterium. *M. tb* can be present in either a latent or active form [1]. Patients with latent forms of TB tend to be asymptomatic, which could happen even if the bacilli are actively replicating in their body [2]. In contrast, active TB patients tend to present with symptoms such as fever, sputum production, hemoptysis, weight loss and decreased appetite [1].

TB is considered one of the top thirteen causes of death globally, and was noted to be the cause of death in 1.5 million people in 2020 [3]. It is estimated that 5–15% of a third of the population who live with latent TB will develop active TB, and that each of these active cases may lead to 10–20 new infections [4]. Upon *M. tb* infection, the host immune system mounts a response against the invading pathogen [5]. Cytokines, such as interferons (IFNs), help recruit the necessary inflammatory cells, including macrophage [6]. This robust and multifactorial response will either result in successful resolution of the infection, latent infection, or tuberculoid disease [5]. Latent TB is a result of granuloma formation that works by creating a barrier around the pathogen, in an effort to protect the host [7].

IFNs are cytokines produced by mammalian cells, and are important for interfering with viral replication and pathogenesis [8]. There are 3 classes of IFNs, which include type I IFN (IFN-α and IFN-β), type II IFN (IFN-γ), and type III IFN (IFN-λ). While type I IFN is induced ubiquitously by innate immune cells, type II is induced by activated immune cells such as T-cells and Natural Killer (NK) cells [9,10]. Type III IFNs are produced by

epithelial cells which protect mucosal surfaces from the attack of pathogens [11]. The pathogenesis of latent TB is still in question. Studies have found that areas of inflammation known as granulomas can be found in both types of infection. While hosts use granulomas to control TB infection, the bacilli itself uses these as a protective mechanism from host antibacterial [12]. Latent TB may have granulomas consisting of T cells, B cells, neutrophils and macrophages, while active TB tend to have more macrophages than lymphocytes [2].

As far as diagnosis goes, certain tests can be done once a patient's medical history is collected and a physical exam is completed. While the Mantoux tuberculin skin test (TST) and blood test for both latent and active TB patients tend to be positive, there are other forms of evaluation such as imaging studies that can be done for further differentiation [1]. Chest radiographs of latent TB patients tend to be clear while those with active infection may show lung lesions which could indicate pulmonary TB. Additionally, acid fast bacilli (AFB) sputum smear is negative for latent TB, and positive for active TB [1].

Studies have found that an increase in type I IFN during TB infection is detrimental, while an increase in type II IFN during TB infection has a protective effect [13,14]. While little is known about the effect of type III IFN on TB infection, studies found it to be a modulator for this pathogen, specifically in the lungs [4]. The aim of this review is to understand the precise roles of interferons on mycobacterial infections, which would allow us to gain insights on protective host immune response against mycobacterial infection.

## 2. Role of Type I IFN (IFN-$\alpha$/$\beta$) on *M. tb* Infection

Type I IFNs are cytokines that influence immune function through inhibition and stimulation of genes in a manner that is rather unpredictable in bacterial infections [15]. In TB patients, type I IFNs activate when the body senses *M. tb* bacterial products via pattern recognition receptors (PRRs), triggering downstream signaling via signal transducers and activators of transcription (STATs), and leading to type I IFN production [16].

As illustrated in Figure 1, type I IFNs, which include IFN-$\alpha$ and IFN-$\beta$, exert their effects by binding to a common receptor that consists of two subunits, interferon $\alpha$ and $\beta$ receptor subunits 1 (IFNAR1) and 2 (IFNAR2), associated with Janus Activated Kinase 1 (JAK1) and tyrosine kinase 2 (TYK2) [17]. Binding of type I IFNs to this receptor leads to dimerization of the aforementioned subunits and autophosphorylation of the receptor associated JAK1 and TYK2, leading to downstream phosphorylation of STAT1 and STAT2 heterodimers [18]. The phosphorylated STAT1 and STAT2 then form a complex with interferon-regulatory factor 9 (IRF-9), which translocates to the nucleus to initiate gene transcription [17–19].

In cases where mycobacterial loads are relatively low, reduced rates of signaling or levels of type I IFN may prime host-protective responses, thus protecting patients against the bacteria [15]. However, as type I IFN increases, as a result of highly virulent strains of *M. tb*, this paradigm shifts. Specifically, type I IFN may upregulate and induce UBE2L6, a gene heavily involved in protein ubiquitination and immune pathways during *M. tb* infection. UBE2L6, through its role in cytokine production and type I IFN pathway induction, may encourage *M. tb* survival; this is supported by an increase in UBE2L6 leading to the inhibition of apoptosis in *M. tb*-infected macrophages [20]. Further, antigen-specific type II IFN expressing CD4 T-cells, which are cells that work to control *M. tb* infection by promoting macrophage antimycobacterial response, may be inhibited by type I IFN [21]. The detrimental influences of type I IFN have been further analyzed within studies in human TB and mouse models of *M. tb* infection.

Mouse studies have shown the reduction in T-helper 1 (Th1) immunity by type I IFN depressing levels of TNF-alpha and IL-12, and decreasing T-cell activation [22]. This decrease in Th1 immunity was associated with lower chances of survival within *M. tb* infected mice populations [23]. Conversely, *M. tb*-infected mice populations had improved life expectancies/outcomes in the absence of type I IFN signaling [23]. A secondary study expressed the significant role of IFNAR1 and STAT2 in macrophage death. Both genes were found to increase mouse survival rates with IFNAR1 cells having decreased type II

IFN and STAT2. This shows that increasing type I IFN signaling can lead to lower mice survival rates [24]. Similarly, another study found that lesser amounts of IFNAR1 present in interferon-γ receptor negative (INFGR-/-) mice greatly reduces survival, supporting the finding that type I IFN signaling plays a role in *M. tb* infections, and that both type I and II define host control of *M. tb*. This shows that type I IFN impairs type II IFN control in mice and humans [25].

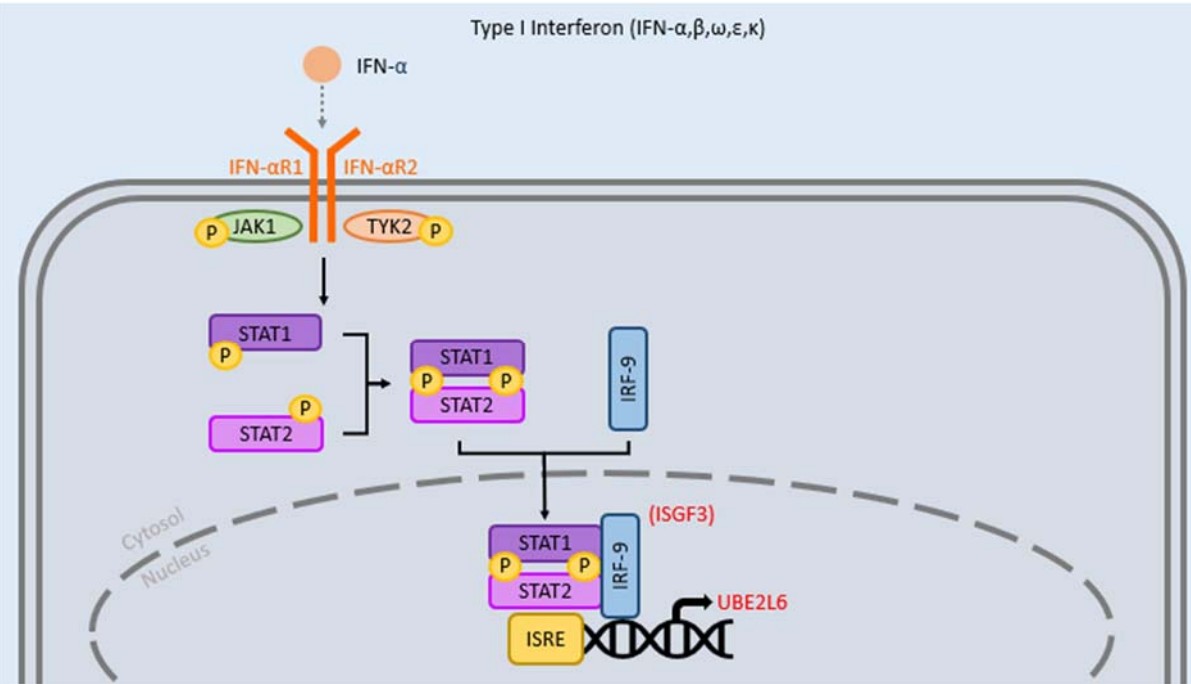

**Figure 1.** Mechanism of action of type I Interferons (IFNs). The receptor is made of two subunits: IFN-αR1 associated with Janus kinase 1 (JAK1) and IFN-αR2 associated with tyrosine kinase 2 (TYK2). Activation results in the phosphorylation of STAT1 and STAT2 which join with IRF-9 to form the Interferon-stimulated gene factor 3 (ISGF3) complex. The complex moves to the nucleus and binds IFN-stimulated response elements (ISRE) to initiate transcription. Type I IFN induces transcription of genes such as UBE2L6, which may encourage *M. tb* survival to the detriment of the host.

Due to the presence of type I IFN, alveolar macrophages that are active in the lungs of patients with *M. tb* undergo early cell death and lead to an accumulation of myeloid cells. Both processes lead to persistent *M. tb* infection and lung inflammation [15]. A study done looking at the gene Il1rn, examined the effects of inducing Il1rn and promoting type I signaling via *M. tb* infected bone marrow in B6 congenic mice. Il1rn encodes for IL-Ra, an adversary to the IL-1 receptor that is responsible for resistance to *M. tb*. The results of this study showed that the mice had higher levels of IL-1Ra protein in their lungs, which failed to fight against the *M. tb* infection due to a block in IL-1 signaling, since IL-1Ra was used instead [24]. This shows that type I IFN has a detrimental interaction of the lung inflammatory process with *M. tb*.

Another study showed a possible mechanism of IFNs invading host *M. tb* cells where serine or threonine kinase is the main driver of type I IFN expression in *M. tb* host cells. Specifically, tank binding kinase 1 (TBK1) triggers the phosphorylation cascade of regulatory factors such as IRF3 and IRF5 that are expressed in *M. tb* [25]. The oxysterol receptor GPR183 was found to be a negative regulator of type I IFN in human monocytes where gene expression of IRF1, IRF5 and IRF7 is upregulated in type 2 diabetic patients who are TB-positive, thus corresponding to the down regulation of GPR183. It was also found that an agonist-induced reaction of GPR183 reduces IFN-β creation. The study found a decrease in IL-10 in *M. tb* infected MNs who took 7α,25-OHC, where IL-10 via type I signaling

prevents the bioavailability of TNF-α, thus increasing *M. tb* growth and preventing *M. tb* phagosomes from growing [26].

Many of these studies illustrate type I IFN as having more of a detrimental role through increasing disease pathogenesis and bacterial expansion when infected with *M. tb*.

### 3. Role of Type II IFN (IFN-γ) on *M. tb* infection

As an integral part of the host innate immune system, type II IFN cytokine, primarily secreted via Th1 cells and NK cells, functions to counteract microbial infection within the host [27–29]. Through binding with cell-surface receptor IFN-γR1 and accessory factor IFN-γR2, type II IFN activates an intracellular signal transduction pathway via JAK1 and JAK2 tyrosine kinases (Figure 2) [30,31]. In doing so, type II IFN plays a variety of roles within the immune system including the promotion of leukocyte-endothelium interactions, antiviral and bactericidal activity, macrophage activation and induction of phagocytosis [28,29].

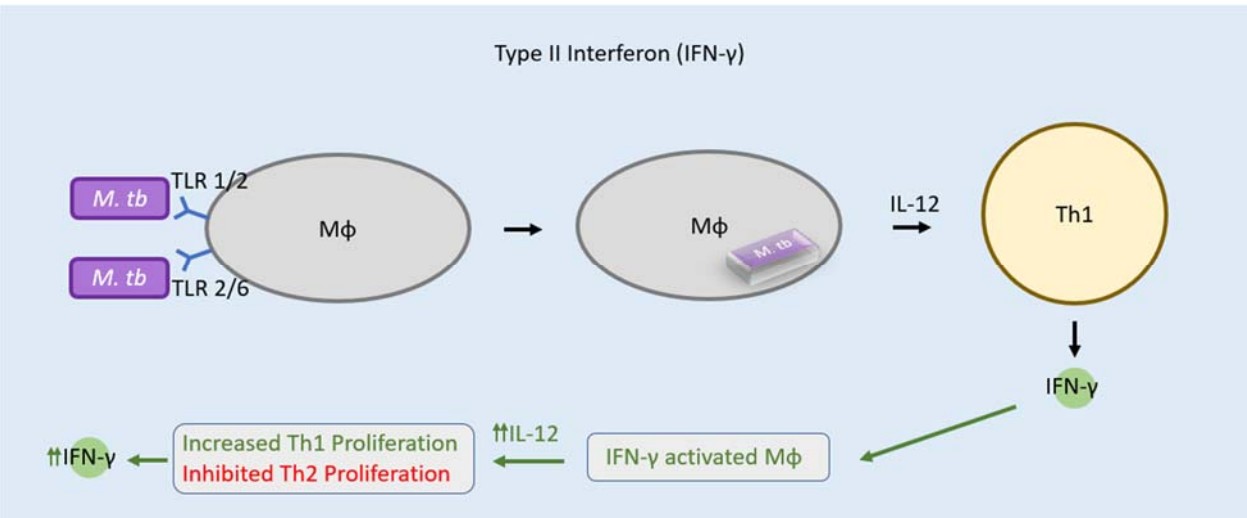

**Figure 2.** Mechanism of action of type 2 IFNs. Type II IFN is the primary mediator which activates macrophages to produce reactive nitrogen species. Macrophages recognize *M. tb* using Toll-like receptors (TLR) and phagocytose the bacteria. Macrophages release IL-12 which drive Th1 cells to release type II IFN. The type II IFN receptor is made of the subunits IFN-*G*R1 and IFN-*G*R2 associated with JAK1 and JAK2, respectively. Activation results in formation of STAT1 homodimers that bind GAS in the nucleus.

When it comes to *M. tb* infections, type II IFN is known to be the primary mediator which activates macrophages necessary for the production of reactive nitrogen species, especially nitric oxide (NO), to restrict the growth of *M. tb* [32]. Upon infection, macrophages recognize *M. tb* via Toll-like receptors (TLR), particularly TLR 1/2 and TLR 2/6, and phagocytose the bacilli. Once phagocytosis occurs, macrophages release IL-12 in order to induce type II IFN production by Th1 cells, facilitating increased macrophage activation and oxidative burst capacity [33]. Macrophages stimulated by IFN-gamma produce greater and continuous amounts of IL-12, further inducing proliferation of Th1 cells to continuously secrete IFN-gamma as well as to inhibit proliferation of Th2 cells which would otherwise block inflammatory cytokines release by Th1 cells [34]. Interestingly, it has been detected that the concentration of IFN-γincreases during latent TB infection (LTBI), suggesting that, in healthy individuals with LTBI, the immune system remains enhanced and active [35,36]. These measurements of IFN-γ have been made possible by electrochemical aptasensors. An aptasensor is a single-stranded DNA probe that can be used to measure the levels of a wide range of targets including amino acids, proteins, and cytokines by specifically binding to, and detecting, their presence and concentration [37,38]. Such modalities are useful in disease diagnosis and environmental bioanalysis, and in the case of *M. tb* infection, allow

for the sensitive quantification of IFN-γ to potentially diagnose LTBI. Defects in innate or adaptive immunity, as well as mutations affecting type II IFN production or signal transduction, have been associated with increased individual susceptibility to disseminated mycobacterial infections [39]. Furthermore, when pathogenic mycobacteria infect host cell macrophages, they block the maturation of the phagolysosomes in order to ensure their own growth, replication, and survival [40].

Macrophages that have been activated by type II IFN have the ability to overcome blockage of phagolysosomes via apoptosis, thereby preventing survival of pathogenic mycobacteria [40]. To demonstrate this, a study conducted by Herbert et al. found that NO, produced by type II IFN-activated macrophages, induced apoptosis of mycobacteria-infected macrophages through caspase activation, via the intrinsic pathway mediated by the production of NO, resulting in mitochondrial outer membrane permeability and release of cytochrome c (Figure 3). Type II IFN induced apoptosis of mycobacteria-infected macrophages in a NO-dependent manner was further confirmed when bone marrow macrophages infected with *M. bovis* Bacille Calmette-Guérin (BCG) or *M. tb*, which were activated by type II IFN, were exposed to caspase 3/7 inhibitors. The results of this experiment show that treatment of these infected macrophages with caspase 3/7 inhibitors rescued intracellular growth of *M. tb* and BCG, demonstrating the fact that NO-induced apoptosis mediated by type II IFN, rather than direct toxicity of mycobacteria by NO, restricts growth of mycobacteria-infected macrophages [40].

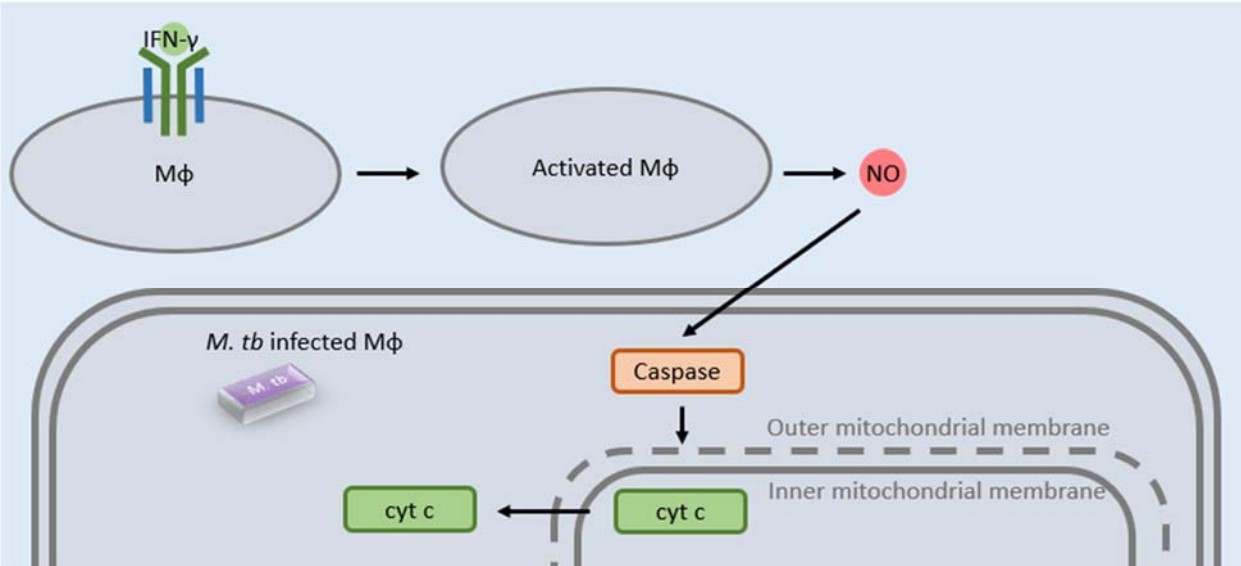

**Figure 3.** Nitric oxide (NO) produced by type II IFN-activated macrophages induce apoptosis of mycobacteria-infected macrophages by increasing outer mitochondrial membrane permeability. This causes leakage of cytochrome c and ultimately cell death.

Similarly, meta-analysis has shown that in comparing a healthy control with a patient group who have TB infection, those in the TB group showed increased levels of type II IFN [41]. This idea that type II IFNs pathway plays a role in restricting *M. tb* infection can be seen in analyzing mendelian susceptibility to mycobacterial disease (MSMD). MSMD is a rare pathology where an individual has inborn errors within varying parts of the type II IFN-dependent immunity pathway [42]. The mutated genes found to result in MSMD included both autosomal (IFNGR1, IFNGR2, STAT1, IL12B, IL12RB1, ISG15, and IRF8) and X-linked (NEMO,CYBB) genes [43]. These individuals are found to have increased susceptibility to infection, including mycobacterial disease [42,44].

## 4. Role of Type III IFN (IFN-λ) on *M. tb* Infection

Type III IFN is also commonly termed IFN-λ; specifically, there are 4 types of IFN-λs. Initially IFN-λ-1, 2, 3 were identified and were discovered as IL–29, 28a, and 28b, respectively. Later on, IFN-λ-4 was discovered, however, it is only expressed in individuals who carry the IFNL4-ΔG gene [45]. These proteins are translated from the genes IFNL1, IFNL2, and IFNL3, which are extensively similar to each other in regard to amino acid identity. The receptor complex, which type III IFN interacts with, is distinct from those of Type I and II IFNs [46,47]. However, interestingly both type I and type III IFNs gene transcription get stimulated in similar manners via pattern recognition receptors [45,48].

The signal transduction of type III IFN relies on specific IL-28Rα chain and IL-10R2 chains. Initially, type III IFN binds to IL-28Rα which leads to a conformational change allowing IL-10R2 to join and form an IL-28Rα-IL-10R2 complex. The intracellular domain of IL-28Rα can then be phosphorylated by tyrosine kinases TYK 2 and JAK 1 [49,50]. Next, STAT proteins identify these phosphorylated regions and form IFN-activated transcription factor 3 (ISGF3), which then moves into the nucleus and binds to interferon-stimulated response elements (ISRE) in the promoter regions of IFN-stimulated genes (ISG) to produce ISG products (Figure 4). This signal transduction pathway is similar to that of type I IFN, however, type III IFN has a lower kinetic profile than type I, but is able to keep a prolonged higher level of ISGs compared to type I [45,51]. In addition, another difference that accounts for the varying responses of type I and type III is that type III IFN receptors are found mostly in epithelial cells in the mucosa [48].

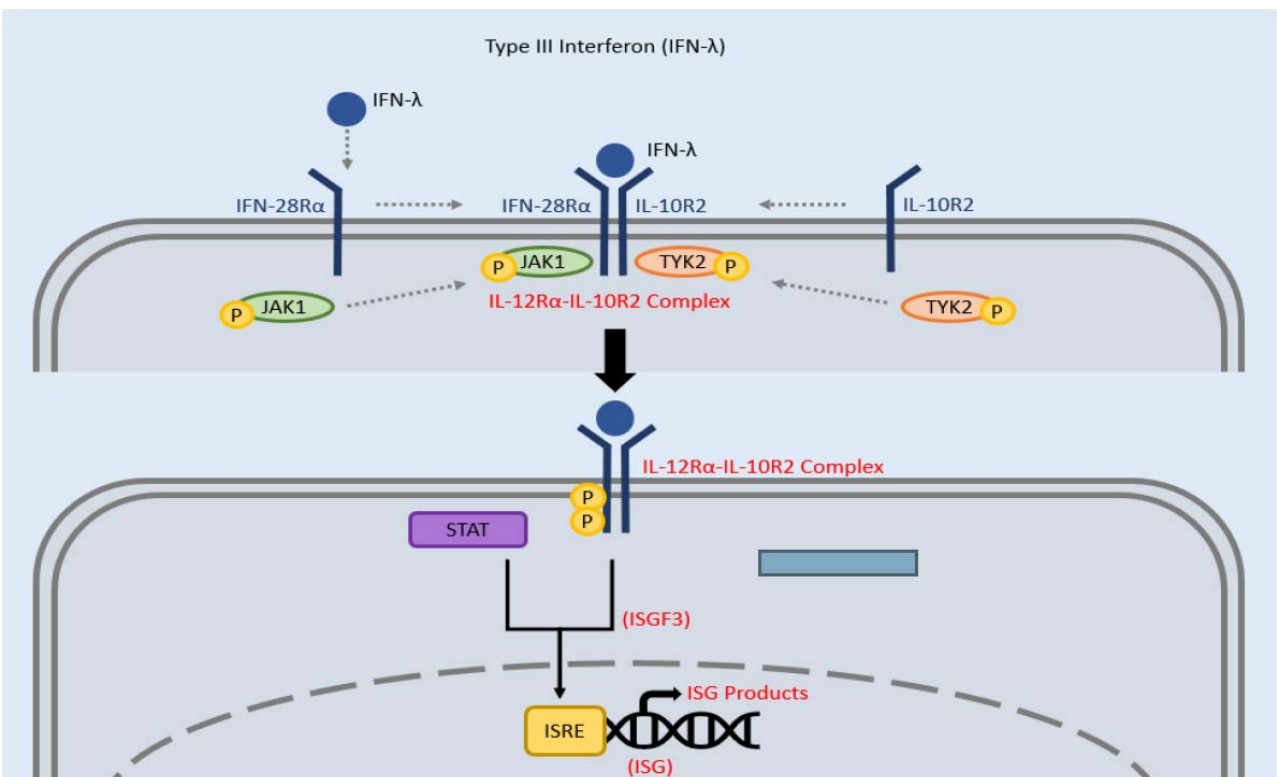

**Figure 4.** Signaling pathway of type III IFN. The receptor is made of two subunits: IFN-28Rα and IL-10R2. Type III IFN binds to the IFN-28Rα receptor which leads to a conformational change, resulting in the addition of IL-10R2. JAK1 and TYK2 then phosphorylate the intracellular domain of IFN-28Rα, allowing STAT to recognize it and bind and form a complex known as ISGF3. This ISGF3 complex moves into the nucleus and bind to the promoter region of ISG, known as ISRE, leading to IFN-stimulated gene products.

While it is known that type III IFN plays a role in antiviral and antitumor pathology, this role is not fully understood in *M. tb* infections. Interestingly, Travar et al. have performed an experiment in sampling the sputum of healthy individuals and those infected with active *M. tb*. They found that there was a statistically significant increase in type III IFNs in those infected with *M. tb*, when compared with healthy individuals [48]. Type III IFN has been postulated to be a modulator of Th1/Th2 response, which are key players in *M. tb* infections [52]. There has been evidence that IFN-λ-2 induces an increase in Th1 response, while IFN-λ-1 inhibits Th2 production [53,54]. It is this change in immune response mediated by type III IFN that may play a minor role in the attempt to clear a *M. tb* infection [48].

## 5. Methods

To find studies for this article on the effects of different IFN classes on the infection clearance caused by *M. tb*, a series of steps were performed. This included collecting data on keywords, inclusion, and exclusion criteria. Information was obtained using Google Scholar, PubMed, Nature, and NCBI databases. Search results included terms such as: "Type I Interferon", "Interferons detrimental to *Mycobacterium tuberculosis*", "Negative effects of Type I interferons", "IFN-γ", "Type 2 interferon", "*Mycobacterium tuberculosis*", "JAK1", "JAK2", "TLR", "Role of IFN-λ", "IFN-λ and *Mycobacterium tuberculosis*", "Functions of IFN-λ within the immune system". Attention was paid in each section to include articles that were relevant and discussed the function of the specific interferon class and its effects on *M. tb*. Exclusion criteria included non-relevance to *M. tb* or a lack of a mechanistic explanation of IFN function.

## 6. Conclusions

Interferons are ubiquitous within the immune response to *M. tb* infection. Type I IFN attenuates the ability to warn off infection through reducing T-cell activation and decreasing important defense cytokines such as TNF-α and IL12. Conversely, type II IFN augments Th-1 signaling, which promotes apoptosis of *M. tb* infected macrophages and decreases the mycobacterium survival in host cells. Still, future studies are required to elucidate the exact mechanism by which type III IFN acts in the immune response in those with tuberculosis. The current literature speculates that type III IFN plays a secondary role in supplementing the Th1 response necessary for host cell containment and resistance to *M. tb*. Levels of each interferon in those with active and latent tuberculosis may suggest varying levels of susceptibility in those who have been exposed to *M. tb*. Future studies should aim to shed light on how the release of IFNs can be influenced to ameliorate preventative initiatives, support host defense, and improve outcomes in those infected with *M. tb*. Certainly, more translational studies using human subjects may provide additional guidance on targeted pharmacology in the treatment of TB, especially in populations with increased susceptibility to *M. tb.* infection.

**Author Contributions:** Conceptualization, V.V.; methodology, D.O., W.N. and N.P.; validation, D.O., W.N. and N.P.; writing—original draft preparation, D.O., W.N., N.P., G.S., S.D., A.P., A.H., D.T., P.V. and K.H.N.; writing—review and editing, D.O., W.N. and N.P.; visualization, D.T. and K.H.N.; supervision, V.V., D.O., W.N. and N.P.; project administration, V.V., D.O., W.N. and N.P. All authors have read and agreed to the published version of the manuscript.

**Funding:** This research received no external funding.

**Institutional Review Board Statement:** Not applicable.

**Informed Consent Statement:** Not applicable.

**Data Availability Statement:** Not applicable.

**Acknowledgments:** We appreciate the funding support from NIH (HL143545-01A1).

**Conflicts of Interest:** The authors declare no conflict of interest.

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
