# Peer review of "Role of Interferons in Mycobacterium tuberculosis Infection"

_clinpract, doi:10.3390/clinpract12050082_

Round 1
Reviewer 1 Report
The manuscript focuses on an interesting subject, the role of interferons in M. tuberculosis infection. I like the manuscript. You may add a few sentences in the introduction to explain which defenses happen at different stages of tuberculosis. Do we have any defense in the latent stage? and also could we differentiate the infected people in the latent and active states via a specific test or the level of INF?
There are minor grammatical and punctuational errors that could be fixed through revision. Here are some examples:
Line 17: plays
Line 43: remove s from protect
Line 62: remove s from genes in genes transcription
Line 78: significant role of
Line 89: that is responsible
Line 115: the host innate immune
Line 119: the promotion of
Line 161: use hyphen INF-activated
Line 170: use is here: with are distinct from those
Line 179: ISG products (plural)
Line 180: a lower kinetic profile than type I but is able
Author Response
Dear Editors and Reviewer#1,
We appreciate the time and effort you have all taken to provide an extensive and thorough review of our manuscript. We are also very encouraged by how well it was accepted and we agree that the suggested edits have greatly improved our manuscript. The objective of our paper was to provide a brief discussion compiling knowledge about subtypes of Interferons and their role in immune defense against M. tb infection. We hope that our revised manuscript based on all of the suggested edits make this objective much more clear, and we look forward to any further suggestion you may have to bring this manuscript to its full potential.
Thank you all for you valuable time.
Reviewer #1:
Comments and Suggestions for Authors
The manuscript focuses on an interesting subject, the role of interferons in M. tuberculosis infection. I like the manuscript. You may add a few sentences in the introduction to explain which defenses happen at different stages of tuberculosis. Do we have any defense in the latent stage? and also could we differentiate the infected people in the latent and active states via a specific test or the level of INF?
Excellent suggestion! Thank you for this input to add additional information about the stages as it provides a more comprehensive insight into the infection process. We attempted our best to add as much information as we can, and you should find this addition in the introduction as well as under subsection about Type II IFN highlighting an interesting correlation between IFN-gamma and the stages of TB as well as its role in diagnostic measures.
There are minor grammatical and punctuational errors that could be fixed through revision. Here are some examples:
Line 17: plays
Line 43: remove s from protect
Line 62: remove s from genes in genes transcription
Line 78: significant role of
Line 89: that is responsible
Line 115: the host innate immune
Line 119: the promotion of
Line 161: use hyphen INF-activated
Line 170: use is here: with are distinct from those
Line 179: ISG products (plural)
Line 180: a lower kinetic profile than type I but is able
Thank you for pointing out these grammatical errors! As much as we attempt to proof-read we seem to miss some minor errors and thus it’s great to receive these comments from a fresh pair of eyes! Excellent job catching these and thank you for directly pointing them out, they have been corrected.
Reviewer 2 Report
Here in this study, the authors have elaborated the therapeutic implications of different types of IFN (IFN I, II, III) against Mycobacterium tuberculosis infections.
It is a well-organized review; however, the figures are elementary and need slight advancements in terms of representation.
Major comment
Though you have highlighted the therapeutic response of IFN-? against M. tb, it would be nice if you could also introduce the diagnostic aspect because that would help connect this study to a broader perspective of therapeutics and diagnostics. IFN-? plays a very important role in discriminating between active and latent tuberculosis (e.g., https://pubmed.ncbi.nlm.nih.gov/35196464/), which is quite beneficial. Mentioning this point will give a comprehensive picture of this interferon against M. tb infection.
Minor comments
The manuscript has certain grammatical errors, which the authors must rectify. I have mentioned a few of them, but not all. The authors should screen the manuscript for such errors.
1) Line 62: There should be a comma before ‘which’
2) Abbreviation used for Mycobacterium tuberculosis should be consistent and italicized at all instances in the manuscript. I have found non-italics versions in numerous instances.
3) Hyphen is missing in certain words, for e.g.
IFN-activated, agonist-induced, IFN-stimulated
4) Line 35: INF should be replaced with IFN, and its full form should be added here as it is the first instance where it is mentioned
5) The full form of the following words must be mentioned at their first use in the manuscript:
STAT, IFNAR, BCG
6) Line 140: Please specify whether it is M. bovis or BCG?
7) M. tb is not written correctly in the figures. Please check all figures.
Author Response
Dear Editors and Reviewer#2,
We appreciate the time and effort you have all taken to provide an extensive and thorough review of our manuscript. We are also very encouraged by how well it was accepted and we agree that the suggested edits have greatly improved our manuscript. The objective of our paper was to provide a brief discussion compiling knowledge about subtypes of Interferons and their role in immune defense against M. tb infection. We hope that our revised manuscript based on all of the suggested edits make this objective much more clear, and we look forward to any further suggestion you may have to bring this manuscript to its full potential.
Thank you all for you valuable time.
Reviewer #2
Comments and Suggestions for Authors
Here in this study, the authors have elaborated the therapeutic implications of different types of IFN (IFN I, II, III) against Mycobacterium tuberculosis infections.
It is a well-organized review; however, the figures are elementary and need slight advancements in terms of representation.
Major comment
Though you have highlighted the therapeutic response of IFN-? against M. tb, it would be nice if you could also introduce the diagnostic aspect because that would help connect this study to a broader perspective of therapeutics and diagnostics. IFN-? plays a very important role in discriminating between active and latent tuberculosis (e.g., https://pubmed.ncbi.nlm.nih.gov/35196464/), which is quite beneficial. Mentioning this point will give a comprehensive picture of this interferon against M. tb infection.
Great suggestion! We have added a general diagnostic topic in the introduction as well as linked the correlation between IFN-gamma and TB in the subsection discussing Type II IFN. I believe it adds great value and allows the paper to be more comprehensive and does indeed provide a broader perspective. In regards to the images, we attempted our best to simplify the mechanistic aspects of the IFNs into an image to clarify it for a broad range of audience. However, we did add an extra line of mechanism to more clearly depict the signaling pathway in Figure 2. In regards to the aesthetics we were limited by the softwares available to us, however if you believe there is a certain aspect of them that should be changed, please let us know as we would delightfully attempt those changes to improve our paper!
Minor comments
The manuscript has certain grammatical errors, which the authors must rectify. I have mentioned a few of them, but not all. The authors should screen the manuscript for such errors.
1) Line 62: There should be a comma before ‘which’
2) Abbreviation used for Mycobacterium tuberculosis should be consistent and italicized at all instances in the manuscript. I have found non-italics versions in numerous instances.
3) Hyphen is missing in certain words, for e.g.
IFN-activated, agonist-induced, IFN-stimulated
4) Line 35: INF should be replaced with IFN, and its full form should be added here as it is the first instance where it is mentioned
5) The full form of the following words must be mentioned at their first use in the manuscript:
STAT, IFNAR, BCG
6) Line 140: Please specify whether it is M. bovis or BCG?
7) M. tb is not written correctly in the figures. Please check all figures.
For comments 1-7, we truly appreciate you pointing out these grammatical errors as they improve the reading of our article. We have corrected these errors and have went through the whole article to ammend other errors as we found them. Excellent catch on these mistakes!